# Measure Once, Mask Once: Delta Refined Block Sparse Attention

## Abstract

Long context inference poses a problem for large language models (LLMs) due to the high cost of quadratic attention with long input lengths. Efficient long context inference is a necessity in order to provide low-cost, low-latency LLM serving endpoints. Sparse attention is one way to mitigate the high cost of long context prefills. Many recent state-of-the-art sparse attention methods can be applied on top of pretrained quadratic transformers without any specific finetuning regimen, however, the main obstacle to overcome when designing sparse attention method lies in deciding which parts to compute and which parts to ignore during the sparse computation. Previous works generally make this decision based on heuristics derived from recurring patterns in the attention matrix or pooled block statistics to select a key-sparse attention mask. We show that these methods result in a suboptimal capture of total attention score mass. In another line of work, key-sparse attention has been shown to induce a distributional shift in attention outputs that can be mitigated by mixing query-sparse attention with existing key-sparse attention masks and combining the outputs. In order to save computation, we propose fusing the query-sparse attention and sparse attention mask generation process, resulting in a novel, dynamic, and query-dependent sparse mask generation. Our method calculates a key-sparse block mask while computing query-sparse attention, and then uses this dynamic attention mask to perform key-sparse attention before combining the outputs. Our method delivers a 2.5x speedup over Flash Attention 3 at 1M tokens and results in a total attention capture which is within 1.5% of the oracle block top-k attention.

## 1 Introduction

Long context processing is a necessary condition for artificial general intelligence. Without long context capabilities, all knowledge would need to be encoded directly into model weights at train time, which is infeasible due to the constantly growing body of knowledge and the long and expensive training times of current LLM foundation models, which can require training on tens of trillions of tokens (Yang et al., 2025a).

Attention (Vaswani et al., 2017) operations come with a quadratic complexity with respect to the input size. In a causal transformer, this means that each token in the sequence must be compared with all previous tokens, leading to a linearly expanding token cache and quadratic computation complexity. This operation poses a problem for long context tasks, which may span millions of tokens. Such scenarios imply a high cost of serving and also high latency for the end user. The expanding memory of a transformer key-value cache (KV cache) is a crucial improvement over recurrent networks (Hochreiter & Schmidhuber, 1997) due to the fact that the cache essentially acts as an expanding memory module. When combined with quadratic attention, this gives a transformer the ability to integrate information over long contexts without ever losing direct access to previously seen tokens. However, attention matrices often come with a high amount of sparsity, which implies that any computation spent on sparse regions of the attention matrix is wasted. However, these sparse tokens may prove to be necessary at a later timestep and therefore should remain in the cache to prevent them from being completely forgotten. Given the quadratic complexity and naturally occurring sparsity, sparse attention mechanisms have become an active and ongoing topic of research (Shah et al., 2024; Lee et al., 2024; Jiang et al., 2024; Willette et al., 2025).

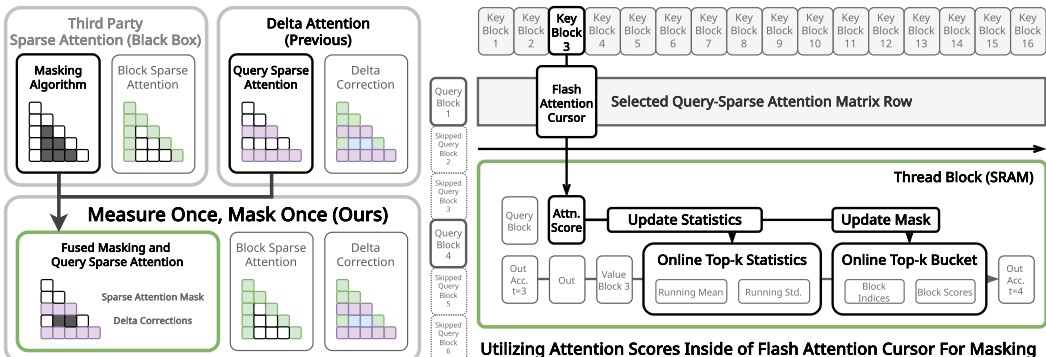

Figure 1: **Concept.** Our "Measure Once, Mask Once" (MOMO) avoids duplicate computation by fusing query sparse attention and sparse attention mask generation into the query sparse attention's flash attention cursor. The key-sparse attention and delta correction are subsequently computed after the fused query-sparse kernel.

Attention sparsity may naturally arise when there is a conditional independence between sections of the input sequence. For instance, consider a synthetic needle-in-a-haystack (NIAH) task or a retrieval augmented generation (RAG) task where documents are conditionally independent from one another. Likewise, in long context natural language tasks, the same situation may arise, as a section of text is unlikely to show high conditional dependence with the entire corpus. In these cases, there should be limited attention between conditionally independent blocks, inducing a large amount of sparsity during the prefill phase. However, in the aforementioned cases, it may be crucial to incorporate fully dense decoding in order to integrate over all the previous tokens in the input when generating a response. Recent work has shown that switching from sparse attention during the prefill to dense attention during decoding induces a distributional shift in the attention outputs, which interferes with the query-key matching during the decoding process (Willette et al., 2025).

The main problem that arises with sparse attention is the need to select which parts of the attention matrix need to be computed and which should be ignored. Although attention matrices have been shown to follow certain patterns, such as vertical lines with diagonal slashes (Jiang et al., 2024), there are still dynamic patterns that may arise since the attention matrix is conditioned on the input. Figure 2b shows such a pattern with an oracle top-k block selection. The slash indices are not constant throughout the attention matrix, and the vertical lines also have a limited and dynamic span. This shows that block-sparse attention should be able to catch dynamic patterns that do not statically extend through the whole attention matrix as fixed vertical and slash indices do. To solve this problem, we propose a novel method of generating a block-sparse attention mask. Building on prior work that combines both key-sparse (KSA) and query-sparse attention (QSA), we devise a two-step process that first performs attention for a sparse query set with a dense key set. While computing the QSA, we collect an online top-k of the block indices that have the highest contribution of attention scores. We then use these collected block indices to compute key-sparse attention and combine the outputs. Our new form of sparse attention results in a 2.5x speedup over the state-of-the-art Flash Attention 3 (Shah et al., 2024) for time-to-first-token (TTFT) on 1M token prefills.

**Our contributions in this work are as follows:**

- We propose *Measure Once, Mask Once* (MOMO), a novel method of dynamic block-sparse attention, which is 2.5x faster than Flash Attention 3 in TTFT on 1M token prefills.
- We show that our dynamic block selection method captures more of the attention mass than baseline block-selection methods and is within 1.5% of an oracle top-k mask.
- We provide three different online top-k block selection algorithms implemented at the kernel level, which achieve $\mathcal{O}(k)$, $\mathcal{O}(\log k)$, and $\mathcal{O}(1)$ time complexity per update, respectively.

## 2 RELATED WORK

MInference (Jiang et al., 2024) found recurring patterns in attention matrices across many common foundation models, discovering that there are reliably recurring patterns, such as the 'vertical and

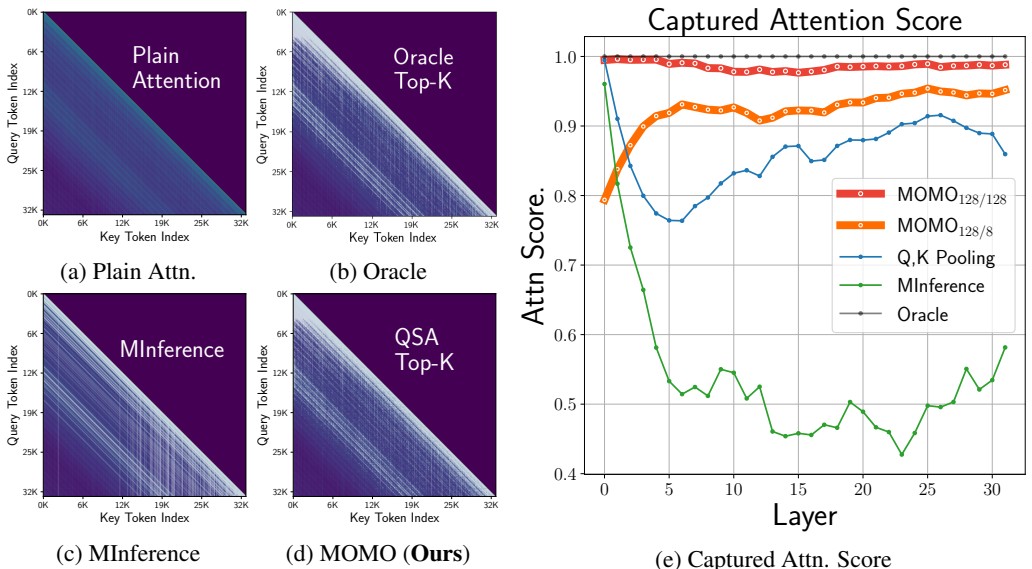

(a) Plain Attn.  (b) Oracle

(c) MInference  (d) MOMO (**Ours**)

(e) Captured Attn. Score

Figure 2: **All attention figures show a 32K attention matrix.** Each pixel represents a single 32x32 block index of the attention matrix. **(d)** During query sparse attention, we track block statistics via an online top-k algorithm. We then use the resulting top-k block indices to generate the key-sparse mask above. Our method captures the dynamic spans of vertical and diagonal slash indices on-the-fly. **(e)** Shows the captured attention mass as compared to the Oracle block mask. All models are limited to selecting 128 total blocks per row. See Fig. 11 for more examples.

slash' patterns that can be seen in Fig. 2. In addition to vertical and slash patterns, they also identify 'Λ-shaped' patterns that are similar to sink tokens and a sliding window (Xiao et al., 2023), and 'block sparse' patterns, which estimate high-scoring blocks before performing sparse attention on the identified blocks. The vast majority of heads utilize the 'vertical and slash' patterns, which are dynamically calculated based on an observation window near the end of the attention matrix. This poses a potential problem, as the observed vertical and slash indices in the final rows of the attention matrix may not extrapolate to the whole extent of the attention matrix as can be seen in the oracle top-k matrix in Fig. 2b. Additionally, while these patterns reliably appear in the current generation of RoPE (Su et al., 2021) based transformers, there is no guarantee that this pattern will continue to appear in future transformers beyond the current generation. For this reason, it is necessary to have a more flexible and dynamic method of generating block-sparse attention masks.

One method used to form block sparse patterns by prior works such as MInference, Seer Attention (Gao et al., 2025), and Sparge Attention (Zhang et al.) involves first pooling blocks of the query and key matrices via a pooling function $\rho : \mathbb{R}^{N \times d} \mapsto \mathbb{R}^{\frac{N}{b} \times d}$ such that the total number of tokens is reduced by a factor of the block size $b$. Sum or mean pooling can both be used for this. After pooling, a compressed attention matrix can be formed $\hat{\boldsymbol{A}} \in \mathbb{R}^{\frac{N}{b_q} \times \frac{N}{b_k}}$. $\hat{\boldsymbol{A}}$ can then be used to generate a block-sparse attention mask due to the fact that within a single arbitrary block index of the queries and keys $B_q(m)$ and $B_k(n)$, the pooling function $\rho$ results in a sum over all dot products in the block.

$$\hat{A}_{mn} = \sum_{i \in B_q(m)} \sum_{j \in B_k(n)} \boldsymbol{Q}_i^\top \boldsymbol{K}_j = \left( \sum_{i \in B_q(m)} \boldsymbol{Q}_i \right)^\top \left( \sum_{j \in B_k(n)} \boldsymbol{K}_j \right) = \rho_m(\boldsymbol{Q})^\top \rho_n(\boldsymbol{K}) \quad (1)$$

A key weakness of this method is that due to relying on the linearity of the sums, it cannot take the exponential of the softmax into account. For example, consider the case where the block may be composed of large positive and negative values such that the total sum is $0$. The large values may dictate that this block is important, however the simple sum has hidden this fact. If the exponential were applied, the large negative values would be squashed to $0$ and the large positive values would not lose their significance. Our method is able to take the exponential into account when computing block scores.

Delta Attention (Willette et al., 2025) identified a distributional shift arising from the use of sparse attention. They found that sparse attention outputs show a significant difference in cosine similarities compared to the outputs of the full attention that the model was trained with. Therefore, if the model were to switch to dense attention during decoding, a common pattern (Jiang et al., 2024; Acharya et al., 2024; Yang et al., 2025b; Yao et al., 2024) in many recent works, there will be a mismatch between sparse/dense attention outputs and therefore a mismatch between queries and keys in later layers in the model. This was shown to lead to a severe performance degradation, as the model will be unable to match appropriate keys for a given query. The authors propose to mitigate this problem by selecting a sparse set of queries $\boldsymbol{Q}_\gamma$ where $\gamma = \{i \iff i \bmod \gamma = 0\}$, and computing full attention with all keys $\boldsymbol{K}$ and values $\boldsymbol{V}$ as $\boldsymbol{O}_\gamma = \sigma(\boldsymbol{Q}_\gamma \boldsymbol{K}^\top)\boldsymbol{V}$. They then compute attention between the full query set $\boldsymbol{Q}$ and a sparse set of keys and values $\boldsymbol{K}_*$, $\boldsymbol{V}_*$ as $\boldsymbol{O}_* = \sigma(\boldsymbol{Q}\boldsymbol{K}_*^\top)\boldsymbol{V}_*$. The sparse sets of keys and values are determined by any generic sparse attention method such as HiP (Lee et al., 2024), MInference (Jiang et al., 2024) or Streaming LLM (Xiao et al., 2023). The difference $\Delta = \boldsymbol{O}_\gamma - (\boldsymbol{O}_*)_\gamma$ is then computed, and this difference $\Delta$ is applied as a corrective term to all queries within the $\gamma$ window,

$$\boldsymbol{O}_i^\Delta = \left[\sigma(\boldsymbol{Q}\boldsymbol{K}_*^\top)\boldsymbol{V}_*\right]_i + \Delta_{\lfloor \frac{i}{\gamma} \rfloor} \tag{2}$$

$$= \left[\sigma(\boldsymbol{Q}\boldsymbol{K}_*^\top)\boldsymbol{V}_*\right]_i + \left[\sigma(\boldsymbol{Q}_\gamma \boldsymbol{K}^\top)\boldsymbol{V}\right]_{\lfloor \frac{i}{\gamma} \rfloor} - \left[\sigma(\boldsymbol{Q}\boldsymbol{K}_*^\top)\boldsymbol{V}_*\right]_{\lfloor \frac{i}{\gamma} \rfloor \gamma} \tag{3}$$

This is done by using two kernel calls, one for the query sparse attention $\boldsymbol{O}_\gamma$, and one for the key sparse attention $\boldsymbol{O}_*$, which uses a pre-existing sparse attention method. While effective, this delta correction is inefficient because the chosen sparse attention method which produces $\boldsymbol{K}_*$ must choose which keys to compute in the sparse attention procedure. However, during the query sparse QSA attention, which is key-dense, all of the keys are scanned for a subset of queries. This means that important block statistics are being computed by the query sparse kernel and then ignored by the sparse attention method, which must use its own internal algorithm to decide which keys are to be computed. Therefore, our key insight is that we can collect useful information about the important blocks during the QSA calculation and return the block indices that are to be used for the key-sparse portion of the attention. This should result in an informed and efficient block sparse attention mask.

## 3 METHOD

Building on the prior work of Delta Attention, we utilize a similar flash attention-based query-sparse-attention kernel. For the query sparsity, we choose queries evenly distributed in a $\gamma$ window such that every $\gamma^{\text{th}}$ query is selected as an input to the QSA kernel. However, as the QSA kernel is scanning full rows of key blocks, we perform an additional online top-k algorithm that collects and stores the identity of the most important key blocks. For an overall algorithm of our method, please see Algorithm 1. We have implemented three different options for online top-k at the kernel level, which allow for different accuracy/latency tradeoffs for top-k block selection (shown in Fig. 4). For each top-k method, we calculate the score for the current block in the same manner. For a given query $\boldsymbol{Q}_i$, and the current key block index $\boldsymbol{K}_{B(j)}$ with block size $\beta$ and block indices $B(j) = \{\beta j, \ldots, \beta(j+1) - 1\}$, we calculate the block score as,

$$S(j) = \log \sum_{l \in B(j)} \exp(\boldsymbol{Q}_i^\top \boldsymbol{K}_l) \tag{4}$$

**Online Exact Top-K.** For exact top-k, we initialize two buffers in the shared memory (SRAM) of the GPU, $\boldsymbol{t} \in \mathbb{N}^k$ for storing the top-k block indices and $\boldsymbol{s} \in \mathbb{R}^k$ for storing the corresponding block scores. The block index buffer is initialized to $\infty$ and the score buffer $\boldsymbol{s}$ is initialized to $-\infty$.

Using the calculated score from Eq. (4), we first select the minimum index of the score buffer $\mu = \arg\min_i \boldsymbol{s}_i$ and update the online top-k scores and indices according to the following update function $U$,

$$U(\mu, S(j)) = \begin{cases} \boldsymbol{s}_\mu > S(j), & \text{pass} \\ \boldsymbol{s}_\mu = S(j), & (\boldsymbol{s}_\mu \leftarrow S(j); \quad \boldsymbol{t}_\mu \leftarrow j) \quad \textit{iff} \ \ j < \boldsymbol{t}_\mu \\ \boldsymbol{s}_\mu < S(j), & \boldsymbol{s}_\mu \leftarrow S(j); \quad \boldsymbol{t}_\mu \leftarrow j \end{cases} \tag{5}$$

In order to avoid unnecessary computation, we perform the argmin over $\boldsymbol{s}$ after the top-k update, which allows us to quickly check whether $\boldsymbol{s}_\mu > S(i)$ on the next iteration and avoid any unnecessary

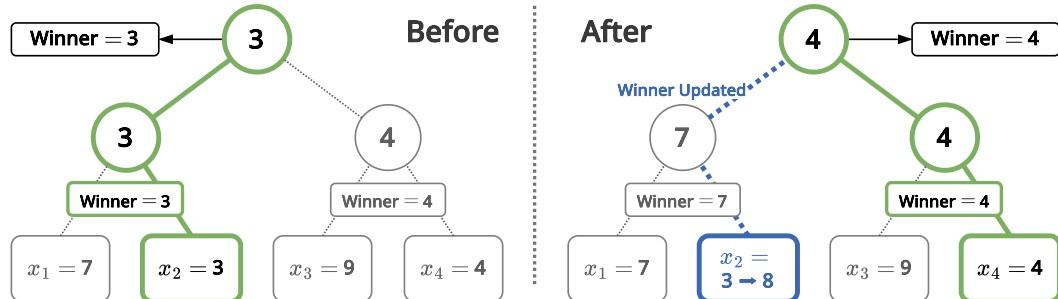

Figure 3: Tournament (winner) tree showing the update process. **(Left):** The current state of the tree at a given timestep. At each depth, the winner (minimum) of the two child nodes are selected until the root contains the global minimum. **(Right):** We receive a score update that is higher than the minimum score in the tree, we must update the tree to properly store this score as one of our top-k scores. We overwrite the leaf node containing the minimum score, and follow the path to the root, storing the new minimum at each depth. The new root contains the global minimum for the tournament tree.

argmin operations over the stored top-k scores and indices buffers $s$ and $t$. However, as $s$ and $t$ are stored in the fast SRAM, this places an inherent limit on the size of the buffers which can be stored and therefore places an upper limit on the values of $k$ which can be used. Therefore, if we are to make use of the more abundant high bandwidth memory (HBM) for the buffers, we must utilize more efficient data structures, as it is impractical to perform $k$ loads/stores for the argmin operation for each block of keys.

**Tournament Tree Exact Top-K.** In the worst case, the above online top-k algorithm requires $\mathcal{O}(k)$ operations at each step due to the $\arg\min$ operation over the score buffer size of $k$ which remains in SRAM. To make use of the more abundant HBM, we may implement the buffer as a partially sorted set using a tournament tree data structure (Knuth, 1998). A tournament tree is a heap-like data structure which can track the minimum value within the tree with an insert complexity of $\mathcal{O}(\log k)$. For this, we need to initialize a buffer of size $2k$ for both $t$ and $s$ as well as one additional buffer $t'$ to store the index of the minimum among the tree leaves. The buffer of $2k$ is needed so that the buffers may be organized into a binary heap-like data structure. Assuming that $k$ is a power of 2, if we consider the root node (minimum value) to be at index location 1 of the $2k$ buffer, and the leaves to be located at the indices $[k, \ldots, 2k-1]$, we may make a comparison between neighboring nodes and store the result at index $\lfloor \frac{i}{2} \rfloor$. Treating the score buffer $s$ as the master buffer, we use the node with the winning score to update block indices $t$ and leaf indices $t'$ with the correct indices corresponding to the winning node. This allows us to set a value of $k$ which is not constrained by the limited amount the SRAM on-chip, with the tradeoff of requiring $\mathcal{O}(\log k)$ loads and stores to HBM per update. As is the case with the online top-k, we can store the minimum scores and indices in SRAM as scalars, and only trigger an update procedure when the current block score is greater than the current minimum. As shown in Fig. 4, the winner tree starts to show lower latency than the online exact top-k when $k > 128$.

**Estimated Online Top-K.** Online exact top-k emits $\mathcal{O}(k)$ complexity per update, and the winner tree emits $\mathcal{O}(\log k)$. Unfortunately, both implementations suffer from expanding compute costs as $k$ increases, as shown in Fig. 4. To this end, we propose a practical $\mathcal{O}(1)$ approximate top-k estimation algorithm as well. For the approximate algorithm, we aim to choose a data-dependent and dynamic acceptance threshold. When a block score exceeds our calculated threshold, we can greedily add it to the indices $t$ and avoid making costly decisions about updating the stored set of indices. This would produce a constant-time insertion complexity. To this end, we assume that the acceptance

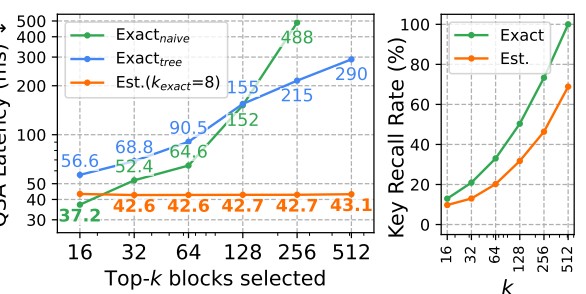

Figure 4: Comparison of online top-k impls.

---

**Algorithm 1** Measure Once, Mask Once (MOMO)

---

**input** Queries $\boldsymbol{Q}$, Keys $\boldsymbol{K}$, Values $\boldsymbol{V}$, Query skip size $\gamma$, Key block size $\beta$, Query block size $\chi$.
1: ▷ *Step 1. Fused Query Sparse Attention & Sparse Mask Selection*
2: **for** $i \in \{0, \gamma, 2\gamma, \dots\}$ **in parallel do**
3:     **for** $j \in \{0, 1, 2, \dots, \lfloor \frac{i+1}{\beta} \rfloor - 1\}$ **do**
4:         $\boldsymbol{O}_\gamma[i/\gamma, B(j)] \leftarrow \sigma(\boldsymbol{Q}_i \boldsymbol{K}_{B(j)}^\top)\boldsymbol{V}_{B(j)}$.
5:         $S(j) \leftarrow \log \sum_{l \in B(j)} \exp\left(\boldsymbol{Q}_i \boldsymbol{K}_l^\top\right)$.
6:         Update $\boldsymbol{t}$, $\boldsymbol{s}$ using $S(j)$ using one of the online top-k algorithms.
7:     **end for**
8: **end for**
9: ▷ *Step 2. Union and trim the block masks to match the block-sparse kernel size $\chi$.*
10: ▷ *For a full specification of trimming, please see Algorithm 2.*
11: **for** $i \in \{0, \chi, 2\chi, \dots\}$ **in parallel do**
12:     Union selected key blocks for the sparse queries in the range $[i, i + \chi)$.
13:     Extract top-$k_{\text{trim}}$ key blocks from the union.
14: **end for**
15: ▷ *Step 3. Compute Block Sparse Attention*
16: $\boldsymbol{O}_* \leftarrow \sigma(\boldsymbol{Q}\boldsymbol{K}_*^\top)\boldsymbol{V}_*$
17: ▷ *Step 4. Apply Delta Correction*
18: $\Delta \leftarrow \boldsymbol{O}_\gamma - (\boldsymbol{O}_*)_\gamma$.
19: $\boldsymbol{O}^\Delta \leftarrow$ Apply $\Delta$ to $\boldsymbol{O}_*$ using Eq. (3).
20: **return** $\boldsymbol{O}^\Delta$.

---

threshold of a single block is proportional to the number of remaining slots divided by the number of remaining blocks in the current row (Eq. (6) RHS). For example, if we have 8 unfilled indices in our top-k buffer $\boldsymbol{t}$ and 32 remaining blocks in the current row, then we should aim to accept $8/32 = 25\%$ of the remaining tokens in the current row. If we assume that $S(j) \sim \mathcal{N}(m(j), \sigma^2(j))$ follows a normal distribution, we can utilize the cumulative distribution function (CDF) to estimate the acceptance threshold (Eq. (6) LHS). For this, we maintain a running mean $m(j)$ and variance $\sigma^2(j)$ of the $S(j)$'s, which can be computed in an online manner (Welford, 1962). While scanning the key blocks, at each step, we pick a threshold $s_{th}(j)$ based on the estimated distribution, and add $S(j)$ to the final list if it exceeds the threshold,

$$\Pr(S(j) > s_{th}(j)) = \frac{\text{remaining\_slots}(j)}{\text{remaining\_key\_blocks}(j)}, \tag{6}$$

where remaining_slots$(j)$ is the number of remaining slots in the $\boldsymbol{t}$ and $\boldsymbol{s}$ top-k buffers, and remaining_key_blocks$(j)$ is the number of key blocks left to scan at the $j^{\text{th}}$ step. Solving Eq. (6), the formula for $s_{th}(j)$ becomes,

$$s_{th}(j) = \sqrt{2} \cdot \sigma(j) \cdot \text{erf}^{-1}(2p - 1) + m(j), \tag{7}$$

$$\text{where } p = 1 - \frac{\text{remaining\_slots}(j)}{\text{remaining\_key\_blocks}(j)}.$$

See Appendix A for a full derivation.

In our approximation, we assume that the dot product scores $\boldsymbol{Q}_i^\top \boldsymbol{K}_l$ are Gaussian distributed due to the fact that the dot product is a sum over a product of random variables. By the central limit theorem (CLT), for a large dimension, these sums tend towards a normal distribution. However, the scores $S(j)$ consist of a $\log \sum_l \exp(\boldsymbol{Q}_i^\top \boldsymbol{K}_l)$ and the inner sum is therefore a sum over log-normally distributed variables. The exact closed form density function for a sum of log-normals $f(x) = \sum_i \exp(\mathcal{N}(m_i, \sigma_i))$ is unknown, however, it is known to be approximated well by another log-normal variable $z = \exp(N(\hat{m}, \hat{\sigma}))$ (Wu et al., 2005). Therefore, the previously mentioned mean and variance updates can be understood to be updating the parameters of this approximate log-normal distribution as,

$$\log \sum_{l \in B(j)} \exp(\boldsymbol{Q}_i^\top \boldsymbol{K}_l) \approx \log z = \log \exp \mathcal{N}(\hat{m}, \hat{\sigma}) = \mathcal{N}(\hat{m}, \hat{\sigma}) \tag{8}$$

In practice, we use a hybrid approach, where we select the top-$k_{\text{exact}}$ key blocks using exact top-k, and the remaining $(k - k_{\text{exact}})$ blocks using the approximate algorithm. This ensures that the most important key blocks are never missed by a bad initial running statistic. Note that the number of exact indices is a constant that does not depend on $k$, thus the insertion complexity remains $\mathcal{O}(1)$.

We compare all three top-k methods in Fig. 4. On the left, we compare the latency of the Query Sparse Attention (QSA) kernel with each top-k method while varying $k$. On the right, we compare what percentage of the attention mass is selected by the QSA kernel relative to the setting $k = k_{\text{exact}} = 512$, with varying values for $k$. For these comparisons, we use a 131K-token context from a RULER needle-in-a-haystack task and a block size of $\beta = 64$. We observe that while the estimated top-k algorithm is less precise (right), its impact on latency stays low due to the $\mathcal{O}(1)$ insert time-complexity as $k$ increases (left). Therefore, using a larger $k$ (e.g. 128 instead of 64) can compensate for the loss of accuracy from using the approximation while maintaining low latency.

**Block Mask Union and Top-k Trimming.** When the QSA kernel has a $\gamma$ value that is smaller than the query block size $\chi$ for block sparse attention, we merge neighboring block sparse mask rows generated by the QSA kernel in order to avoid ambiguity. However, simply taking the union of the candidate blocks in each row may result in a larger-than-expected attention mask. Therefore, we trim the attention

Table 1: Effect of Trimming Alg.

| Method | RULER ($T$=128K) | Latency (ms) |
|---|---|---|
| FA3 | 77.92 | 31.5 (-) |
| Union w/o Trim | 77.57 | 45.0 ($-30\%$) |
| Union w/ Trim | 77.20 | 26.5 ($+19\%$) |

mask by discarding all but the top-$k_{\text{trim}}$ blocks from the union of the candidate blocks in each row, where $k_{\text{trim}}$ is an adjustable hyperparameter. These blocks are ranked by their block scores (Eq. (4)), which are already available from the aggregated statistics gathered from the QSA kernel (see Fig. 1). The complexity of this merging process, which includes the union, mean calculation, and sorting, is $O(k \log k)$, a full algorithm can be found in Algorithm 2. As shown in Table 1, pruning unimportant blocks in this manner significantly improves latency of a single attention layer over a simple union.

## 4 EXPERIMENTS

**Experiment Settings.** For all experiments we use a single node equipped with 8x H100 GPUs. To assess the effective context length of each attention method, we use the RULER benchmark (Hsieh et al., 2024). As a synthetic "needle-in-a-haystack" benchmark, RULER is ideal for evaluating an attention mechanism's ability to accurately retrieve information. For real-world language tasks, we use the English subset of InfiniteBench (Zhang et al., 2024), which features question-answering and summarization tasks derived from long book passages. We evaluate summarization using the ROUGE-L score. For question answering (QA), we report two metrics: F1 and recall. The F1 score measures the harmonic mean of precision and recall between the predicted and ground-truth answers. Recall measures whether the ground-truth answer is contained within the model's generated output. For tasks that do not require summarization or QA, we report the accuracy metric for InfiniteBench. For our method, a subscript of $64/64$ indicates $k = 64$ and all 64 are exact top-k and a subscript of $128/8$ means $k = 128$ and only 8 are exact top-k. The remaining 120 will utilize the approximate top-k algorithm outlined in Section 3.

**Baselines.** We use **Flash Attention 3** (Shah et al., 2024) as our efficient dense attention baseline. We use the SGlang open-source LLM serving framework (Zheng et al., 2024) for our method and all of the following baselines. For sparse attention, we compare against four baselines: (1) **Minference** (Jiang et al., 2024), which uses either a vertical-slash patterns or a block sparse pattern derived from mean-pooling queries and keys before estimating block attention scores; (2) **Delta Attention** (Willette et al., 2025), which uses InfiniteHiP for sparse attention and applies a correction with $\gamma = 16$; (3) **InfiniteHiP** (Lee et al., 2025), which employs a multi-stage hierarchical pruning algorithm; and (4) **Mean-Pooling Sparse Attention** (Jiang et al., 2024), a component of both Minference and Seer Attention that uses block sparsity based on mean-pooled attention score pruning. We include mean-pooling as a representative baseline because it is a common and efficient training-free mechanism for attention sparsification used in prior work (Jiang et al., 2024; Gao et al., 2025; Lai et al., 2025). For MInference baselines, we utilize the official published library for RULER tasks, and use SGLang's Dual Chunk Flash Attention, which has the MInference CUDA kernels merged directly into SGLang for all other experiments and ablations. Our method, as well as Delta and HiP attention variants rely on Triton (Tillet et al., 2019) kernels instead of CUDA kernels.

| Method | 131K | 65K | 32K | 16K | 8K | 4K | **Avg.** |
|---|---|---|---|---|---|---|---|
| FA3 | 78.02 | 86.50 | 89.54 | 93.63 | 94.43 | 96.26 | 89.73 |
| MInference | 73.16 | 84.25 | 90.22 | 94.28 | 93.86 | 95.49 | 88.55 |
| InfHiP | 74.99 | 84.38 | 88.17 | 93.66 | 94.43 | 96.24 | 88.65 |
| InfHiP + $\Delta$ | 77.37 | 86.34 | 89.74 | 93.50 | 94.47 | 96.26 | 89.61 |
| Mean Pool | 15.57 | 32.48 | 37.64 | 54.17 | 81.56 | 96.24 | 52.95 |
| **MOMO**$_{64/64}$ | 75.79 | 86.09 | 89.54 | 93.80 | 94.41 | 96.31 | 89.33 |
| **MOMO**$_{128/8}$ | 77.20 | 85.33 | 89.75 | 93.84 | 94.46 | 96.31 | 89.49 |

Table 2: **RULER on Llama 3.1 8B.** Our MOMO shows a notable increase in performance over most sparse attention methods at the longest context lengths, while the performance of all sparse attention methods saturates for context lengths less than 65K.

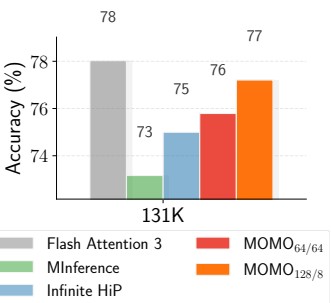

Figure 5: RULER 131K (the longest subset) shows the largest improvement with our method.

| Method | P.K | Num. | KV | Sum | Choice | QA-F1 | QA-R | M.F | Avg. |
|---|---|---|---|---|---|---|---|---|---|
| FA3 | 100 | 100 | 100 | 32.19 | 88.21 | 24.41 | 58.33 | 43.37 | 68.31 |
| Minference | 100 | 100 | 100 | 33.94 | 86.46 | 24.55 | 59.81 | 34.86 | 67.45 |
| InfHiP | 100 | 100 | 98.00 | 33.59 | 78.17 | 25.96 | 54.72 | 36.57 | 65.88 |
| InfHiP + $\Delta$ | 100 | 100 | 98.00 | 33.20 | 81.66 | 23.52 | 57.24 | 36.29 | 66.24 |
| Mean-Pool | 100 | 100 | 0.20 | 18.01 | 69.87 | 11.53 | 23.20 | 36.57 | 44.92 |
| **MOMO**$_{64/64}$ | 100 | 100 | 100 | 32.85 | 83.41 | 22.64 | 57.67 | 36.00 | 66.57 |
| **MOMO**$_{128/8}$ | 100 | 100 | 100 | 33.42 | 83.84 | 22.35 | 56.14 | 36.00 | 66.47 |

Table 3: **InfiniteBench on Qwen3 30B A3B** Our method delivers competitive performance which is within %1 of the best baseline score on average. When considering both performance and latency (Fig. 10), our method is on the Pareto frontier in terms of the performance vs latency tradeoff.

## 4.1 BENCHMARKS

**RULER.** In Table 2, we evaluate our MOMO on the RULER benchmark with the Llama 3.1 8B Instruct model (Llama Team, 2024) and compare with baseline methods. Compared to the baseline sparse attention methods, MOMO achieves scores only down 0.24%p from the dense FlashAttention3 on average. Notably, our method shows a greater increase in accuracy at the longest context lengths (65K and 131K), whereas the performance of all methods saturates at context lengths less than 65K. As shown in Fig. 2, our method is able to capture more dynamic patterns in the attention matrix which include dynamic spans of vertical and slash patterns where the lines do not continue throughout the entire attention matrix. As context grows longer (Fig. 2e), this results in a captured attention score which is much closer to an oracle than baseline methods.

**InfiniteBench.** In Table 3, we evaluate MOMO on the InfiniteBench benchmark ($T$=256K) with Qwen3-30B-A3B-2507 model (Yang et al., 2025a). Our model delivers competitive performance, which is above that of all baselines except MInference, which shows performance within %1 of ours. However, we note that the pre-tuned MInference configurations lead to higher latency than FA3 for 256K context lengths which places our method on the Pareto frontier in terms of performance vs. latency (see Fig. 10).

**Attention Token-wise Efficiency** In Table 4, we present the correlation between model performance and computational efficiency, exploring both theoretical metrics (the number of attended tokens) and practical benchmarks (end-to-end speed). $k_{\text{sparse}}$ denotes how many tokens are attended sparsely (through block sparse attention), and $k_{\text{dense}}$ denotes

| Method | $k_{\text{sparse}} \times 10^3$ | $k_{\text{dense}} \times 10^3$ | E2E (s) | $\infty$Bench |
|---|---|---|---|---|
| FA3 | 0.0 | 1048.57 | 76.92 | 68.31 |
| MInf. | 37.38 | 0.0 | 80.32 | 67.45 |
| InfHiP | 2.88 | 65.53 | 34.90 | 65.88 |
| InfHiP+$\Delta$ | 2.88 | 126.97 | 39.88 | 66.24 |
| **Ours** | 3.84 | 126.97 | 33.54 | 66.64 |

Table 4: Number of attended attention key tokens per query at $T$=1M on Qwen3 30B A3B.

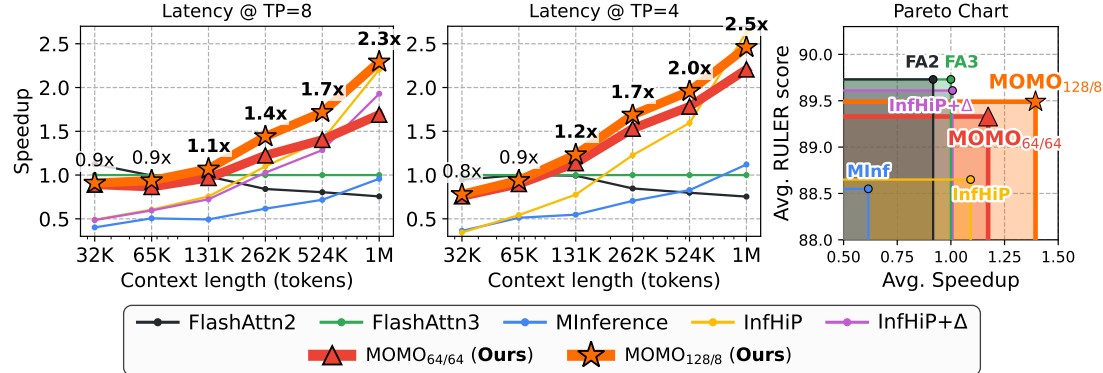

Figure 6: **Real-world end-to-end prefill latency (TTFT) speedup relative to Flash Attention 3.** We measure the end-to-end prefill latency on 8x H100 (left) and 4x H100 (middle) using the Qwen3 30B model on the SGLang inference framework. On the right, we plot a Pareto chart showing the trade-off between accuracy (RULER score) and average of speedup over the 32K – 1M range.

how many tokens are attended densely (through query sparse attention and flash attention). Interestingly, the total number of attended tokens $k_{\text{sparse}} + k_{\text{dense}}$ does not directly indicate end-to-end latency. Instead, $k_{\text{sparse}}$ dominates the latency due to there being fewer opportunities for efficient cache usage or hardware acceleration compared to dense attention. Due to the dual usage of query-sparse and key-sparse attention in our method, we are able to strike a balance between $k_{\text{sparse}}$ and $k_{\text{dense}}$ in order to achieve better efficiency.

**End-to-end latency.** In Fig. 6, we compare the end-to-end prefill latency of our method against baselines using the SGLang serving framework (Zheng et al., 2024), using the Qwen3 30B A3B model (Yang et al., 2025a) which has a built-in pre-tuned MInference configuration. Relative to the state-of-the-art dense attention method Flash Attention 3 (FA3), our MOMO achieves an up to 2.3x speedup using 8x H100 GPUs, and up to 2.5x speedup using 4x H100 GPUs using tensor parallel. In contrast, MInference narrowly beats FA3 by a small margin at 1M tokens, and is slower than FA3 in shorter context lengths. While InfiniteHiP manages to beat our MOMO at 1M tokens, it is slower than ours in the <1M token setting. Our method is the only one that consistently outperforms FA3 in the 131K – 1M range. Combined with the strong results in RULER and InfiniteBench benchmarks, this demonstrates our method's practical usefulness for speeding up inference in real-world LLM serving, also illustrated by the Pareto chart in Fig. 6 (right).

## 4.2 ABLATION

**Estimated vs. Exact $k$.** In Section 3, we outlined two exact top-k methods (online top-k and winner tree) and one approximate top-k method. When using the approximate top-k, in practice, we set a small exact online top-k buffer and utilize the approximate top-k method for the remaining $k_{\text{est}} = k - k_{\text{exact}}$ slots in the buffer. This opens up the possibility for different settings for $k$ and $k_{\text{exact}}$ within a given budget for $k$. To investigate the effect of these settings, we perform an abla-

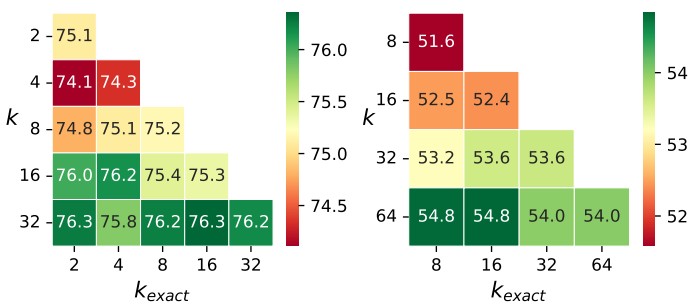

(a) RULER (Llama 3.1 8B)  (b) Inf.Bench (Qwen3 30B A3B)

Figure 7: A heatmap showing the relationship between observed K and exact K values. As total K increases, the overall performance of MOMO increases in turn.

tion shown in Fig. 7 by picking a spectrum of values for $k_{\text{est}}$ given a total $k$ budget. We perform this experiment on both RULER (131K only) with Llama 3.1 8B and on InfiniteBench (`mc.QA` subset) with Qwen3 30B A3B. For both settings, we see increasing performance with an increasing setting for $k$. In contrast, we observe that increasing $k_{\text{exact}}$ while holding $k$ fixed has little effect on the

performance, which suggests that the approximate top-k algorithm is sufficiently accurate that the performance is saturated with small $k_{\text{exact}}$.

**Sparsity By Context Length.** In Fig. 8, we look at the overall sparsity as a function of the context length up to 1M tokens. With $K$ as the number of blocks computed with a block size of $B_k$, $W$ as window tokens, $S$ as sink tokens, and $C$ representing the context length, the formula to calculate the overall sparsity $\zeta$ is given by,

$$\zeta = 1 - \min\left(1, \frac{B_k K + S + W + \frac{C}{\gamma}}{C}\right) \quad (9)$$

This takes all components into account and amortizes the cost of the delta correction over the $\gamma$ rows where the same delta will be repeated. As $C$ grows larger, the effect of the constant terms diminish which places the upper bound of the sparsity $\overline{\zeta}$ at $\overline{\zeta} = 1 - \frac{C}{\gamma}$ due to the query sparse kernel.

**Ablation of $\gamma, \chi$.** In Fig. 9, we show the interactions for $\gamma$ which controls query sparsity in the QSA kernel and $\chi$ which controls the query block size in the block sparse attention. Setting $\gamma$ to a higher value will cause more overall sparsity and also degrade the accuracy of the attention mask as we are applying the mask from row $i$ to a larger number of subsequent rows $j \in [i, \ldots, i + \gamma]$. On the other hand, setting $\chi$ to a value much larger than $\gamma$ will cause more rows to be condensed in the union and trimming (Algorithm 2) and will likewise degrade the precision of each predicted mask. Figure 9 shows this interaction on the RULER 131K Avg. benchmark for Llama 3.1 8B Instruct. We find that the average performance decreases as the overall sparsity ($\gamma$) increases and also that as $\chi$ increases over $\gamma$ the performance tends to be decrease. We chose the settings of $\gamma = 16, \chi = 128$ in our experiments as sensible defaults due to the common setting of 128 as the query block size of flash attention kernels.

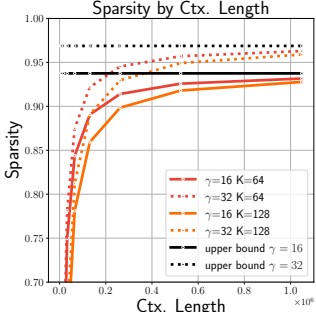

Figure 8: Sparsity calculated as a function of context length.

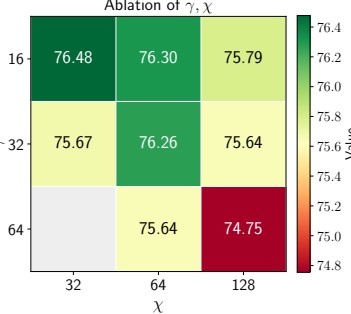

Figure 9: Ablation of the query sparse parameter $\gamma$ and the block-sparse attention query block size $\chi$.

## 5 LIMITATIONS & FUTURE WORK

Our method uses a query sparse kernel which densely scans entire rows of keys in order to collect the block statistics needed for generating a sparse attention mask. This opportunity to collect block statistics comes as a side effect of the query sparse kernel that is needed for the delta correction term outlined in Eq. (3). This means that while fast and efficient, the overall attention algorithm remains quadratic since the query dimension is reduced by a factor of $\gamma$. This limitation could be further mitigated if the key-dense delta correction could be approximated by a pooled set of key-value vectors, then the second dimension could likewise be reduced by another factor in the key dimension which would have a multiplicative effect on the total amount of computation reduction. Furthermore, to gain an even better approximation, one might consider hierarchically pooling key-value blocks in order to achieve an even better approximation to the key-dense delta correction.

## 6 CONCLUSION

In this work, we introduced *Measure Once, Mask Once* (MOMO), a novel method for efficient long-context inference that fuses query-sparse attention with dynamic block mask generation before performing key-sparse attention followed by a delta correction. By leveraging information already computed during query-sparse attention, our approach eliminates redundant computation and produces a key-sparse mask that dynamically adapts to the input. This design allows us to capture nearly the full attention mass of an oracle top-$k$ mask, while achieving up to a 2.5x speedup over FlashAttention3 at the million-token scale. Across RULER and InfiniteBench, our method consistently approaches dense attention in accuracy while delivering lower latency, making it practical for real-world long-context inference.

**Reproducibility Statement** To aid in experiment reproducibility, we have based all of our experiments on publicly available datasets. We have included a detailed algorithm in Algorithm 1, and have explained all relevant hyperparameters in Section 4. We have included the kernel level code for our method in the supplementary file, which contains all of the newly proposed components of this work. Additionally, we will make a full open-source release of our method upon acceptance.

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

---

**Algorithm 2** Trimming Algorithm

---

**input** BSA scores $s$, BSA indices $t$, Num target blocks $K$, Query skip size $\gamma$, Key block size $\beta$, Query block size $\chi$.
1: ▷ *Step 1. Calculate number of union rows $U$, reshape, and flatten $s, t$*
2: $U \leftarrow \frac{\chi}{\gamma}$.
3: $s, t \leftarrow$ reshape $s, t$ from $(B, H, S, K) \rightarrow (B * H, \frac{S}{U}, U * K)$
4: ▷ *Step 2. Sort scores and indices by indices sorting idx in last dimension*
5: $s, t \leftarrow$ sort $s, t$ according to argsort$(t)$ indices
6: ▷ *Step 3. Take mean of scores which have repeat indices in last dimension*
7: $s, t \leftarrow$ reduce $s, t$ by mean where $t$ has repeated indices.
8: ▷ *Step 4. Final sort and top-k.*
9: $s, t \leftarrow$ sort $s, t$ according to argsort$(s)$ in descending order.
10: $t \leftarrow t[..., :\text{K}]$
11: **return** $t$.

---

## A  THRESHOLD FORMULA DERIVATION

Restating the equation,

$$\Pr(S(j) > s_{th}(j)) = \frac{\text{remaining\_slots}(j)}{\text{remaining\_key\_blocks}(j)}. \tag{6}$$

Substituting using $p = 1 - \frac{\text{remaining\_slots}(j)}{\text{remaining\_key\_blocks}(j)}$,

$$\Pr(S(j) > s_{th}(j)) = 1 - p. \tag{10}$$

Since we assumed $S(j) \sim \mathcal{N}(m(j), \sigma^2(j))$, $\frac{S(j) - m(j)}{\sigma(j)} = X$ where $X \sim \mathcal{N}(0, 1)$. Thus

$$1 - p = \Pr(S(j) > s_{th}(j)) \tag{11}$$

$$= \Pr\left(X > \frac{s_{th}(j) - m(j)}{\sigma(j)}\right) \tag{12}$$

$$= 1 - F_X\left(\frac{s_{th}(j) - m(j)}{\sigma(j)}\right), \tag{13}$$

where $F_X(x) = \frac{1}{2}\left[1 + \text{erf}(x/\sqrt{2})\right]$ is the cdf of $X$. Thus

$$1 - p = 1 - F_X\left(\frac{s_{th}(j) - m(j)}{\sigma(j)}\right) \tag{14}$$

$$= 1 - \frac{1}{2}\left[1 + \text{erf}\left(\frac{s_{th}(j) - m(j)}{\sigma(j)\sqrt{2}}\right)\right], \tag{15}$$

Solving for $s_{th}(j)$, we arrive at the result

$$\Rightarrow 2p - 1 = \text{erf}\left(\frac{s_{th}(j) - m(j)}{\sigma(j)\sqrt{2}}\right) \tag{16}$$

$$\Rightarrow s_{th}(j) = \sqrt{2} \cdot \sigma(j) \cdot \text{erf}^{-1}(2p - 1) + m(j). \tag{17}$$

## B  TRIMMING ALGORITHM

As noted in Section 3, the QSA kernel $\gamma$ parameter and the block sparse attention kernel's query stride setting may not be the same which created the need for an algorithm to merge rows of our target BSA indices in order to keep the number of computed blocks in line with the targeted number of blocks we wish to compute.

## C   Extra Efficiency-Accuracy Tradeoff Charts

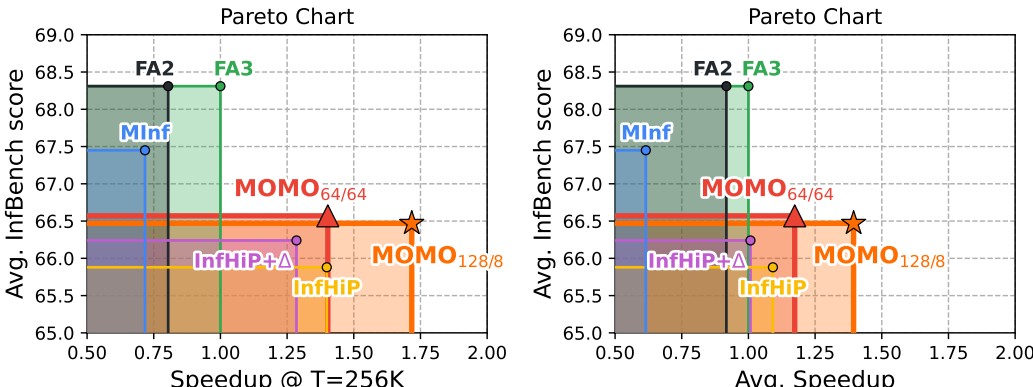

Figure 10: **Efficiency-Accuracy (InfiniteBench) Tradeoff Charts.** The speedup is relative to Flash Attention 3, measured on 8x H100 with Qwen3 30B model.

## D   Extra Attention Matrix Plots

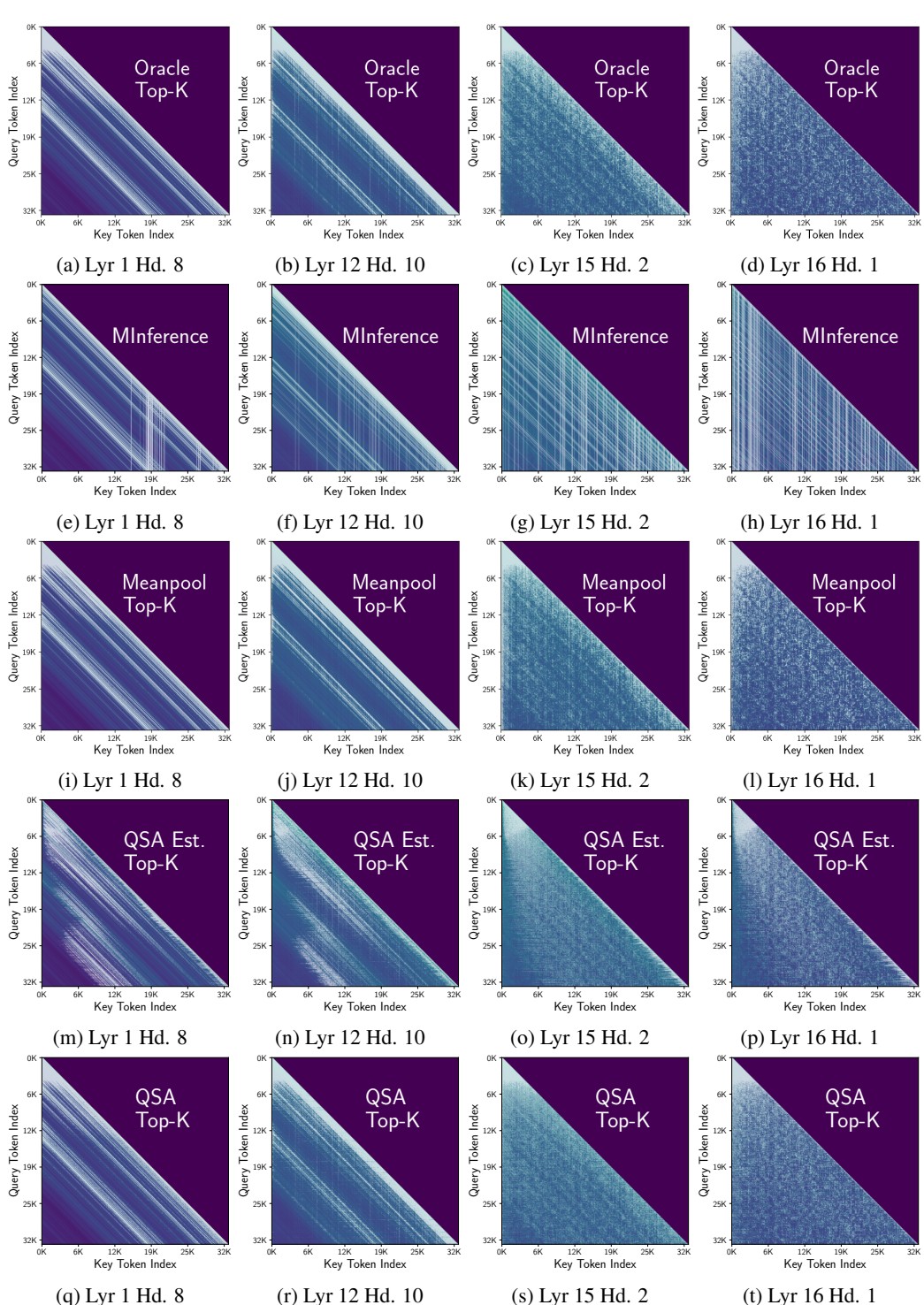

Figure 11: Extra examples of attention figures from different sparse attention methods.

# E    LLM USAGE

In preparing this work, we utilized publicly available LLM's for the following purposes:

- Finetuning some sections of writing after writing a complete draft.
- Generating boilerplate code for some verbose kernel operations, which was then modified to fit our needs.

