# OpenReview forum: "Measure Once, Mask Once: Delta Refined Block Sparse Attention"
_ICLR.cc/2026/Conference — Submitted to ICLR 2026_

### Official Review · Reviewer_yjfS · 2025-10-18

**Soundness:** 2
**Presentation:** 2
**Contribution:** 1
**Rating:** 2
**Confidence:** 3

**Summary:**

This paper proposes Measure Once, Mask Once (MOMO), a method for efficient long-context inference in large language models. MOMO builds on Delta Attention by fusing query-sparse attention (QSA) and sparse mask generation into a single pass. During QSA, the method collects block-level attention statistics via an online top-k algorithm to determine which key blocks to retain in the subsequent key-sparse attention (KSA) step. The fused process aims to eliminate redundant computation from Delta Attention, which previously required a separate mask-selection stage. The authors implement several kernel-level online top-k algorithms (exact, tournament-tree, and approximate) and demonstrate up to 2.5× speedup over FlashAttention-3 at 1M-token contexts

**Strengths:**

1. Introduce and solidly implement three online top-k query sparse attention block selection algorithm.
2. The acceleration gain is good.

**Weaknesses:**

1. The proposed method primarily reuses the Delta Attention pipeline (query-sparse + key-sparse + delta correction). The only new component is the reuse of QSA statistics to select key blocks, i.e., performing mask generation during QSA rather than afterward. This represents a minor engineering optimization rather than a substantive conceptual advance in attention modeling or theory.
2. The paper does not provide theoretical justification or formal analysis showing why the online top-k selection during QSA approximates the oracle mask or preserves attention mass. From the evaluation, the original delta attention can also achieve a comparable accurate performance. It seems the major advance of this paper is to reduce the overhead by reusing the QSA statistics to generate mask.
3. Lack discussion or approaches to determine how to combine or set the number of k and k_exact.
4. Critical hyperparameters (γ for query skip size, χ for block size) are not explored. These are likely to affect both performance and accuracy, yet are fixed without justification.
5. Detailed latency breakdown is helpful for understanding the overhead and source of acceleration.
6. Algorithm 1 is listed but not discussed.

**Questions:**

See weakness.

---

> ### Author Response · Authors · 2025-11-21
> **Response (Part I)**
>
> Thank you for your time and effort in reviewing our work and noting the strength of our implementation and acceleration. We will respond to each of your comments below.
>
> ---
>
> > W1: [...] primarily reuses the Delta Attention pipeline (query-sparse + key-sparse + delta correction). The only new component is the reuse of QSA statistics to select key blocks [...]. This represents a minor engineering optimization rather than a substantive conceptual advance in attention modeling or theory.
>
> We respectfully disagree with this comment. **Our work proposes a completely novel block-sparse attention algorithm which captures more attention mass than current SOTA methods like MInference, while maintaining better latency and achieving similar or better performance.**
>
> While we do use components adapted from the Delta Attention pipeline, **our proposed sparse attention is unique, novel, and powerful.** To claim that a novel sparse attention mask generation method is merely "minor engineering" and not substantive, this logic would also apply directly to the MInference and HiP sparse attention frameworks as well which are analogous sparse attention mask generation works.
>
> ---
>
> > W2: The paper does not provide theoretical justification or formal analysis showing why the online top-k selection during QSA approximates the oracle mask or preserves attention mass.
>
> The empirical findings from Delta Attention showed that the difference between sparse and dense attention outputs could be reused to fill in the "missing attention" of the sparse attention outputs of nearby tokens. **The fact that this has been shown to be effective is empirical evidence in support of our method.** If attention from one row can be re-used in subsequent rows, this means that the attention patterns in those subsequent rows is similar enough to mimic the original missing attention. **Therefore, when we calculate our attention mask based on one row, we transitively apply this logic of the row-similarity to the mask generation process.**
>
> **In short, delta attention implies that an oracle mask generation based on one row can be extended to subsequent rows as well.**
>
> ---
>
> > W2: [...] the original delta attention can also achieve a comparable accurate performance. It seems the major advance of this paper is to reduce the overhead by reusing the QSA statistics to generate mask.
>
> **You are correct in noting that the major advance of this paper is to use QSA to generate a sparse attention mask in a more efficient and effective way.** The original delta attention needed to use the QSA kernel along with an existing sparse attention framework (like MInference or HiP). Our major advancement lies in collecting information in the QSA kernel which was overlooked and allows us to generate a more accurate attention mask (quantified by attention score capture) in a more efficient way (quantified by latency).
>
> ---
>
> > W3: Lack discussion or approaches to determine how to combine or set the number of k and k_exact.
>
> Figure 7 shows k has a larger impact on the model accuracy than k_exact, and the performance impact of varying k_exact is negligible. Therefore, we set k_exact to a small number (8 of 128 total k) when using the approximate top-k algorithm.
>
> Figure 4 shows that when k_exact is fixed, the overall k can be increased with no effect on top-k latency. Therefore, we conclude that it is best to us can be set to a fairly high value (k_exact=8, k=128).
>
> ---
>
> > W5: Detailed latency breakdown is helpful for understanding the overhead and source of acceleration.
>
> In Table 4 and the surrounding lines, we showed that sparse attention increases latency overhead per attended token compared to dense attention: When the same number of tokens are attended, it is more efficient to use dense attention rather than sparse attention.
>
> This is the source of acceleration in our method. Utilizing query sparse attention (QSA), which effectively has the same performance overhead as dense attention, allows MOMO to lower the dependence on sparse attention and balance between the two. Prior sparse attention methods need more of the inefficient sparse attention because they do not utilize QSA.
>
> ---
>
> > W6: Algorithm 1 is listed but not discussed.
>
> We have added a reference to algorithm 1 in the first paragraph of the method section of our paper. We have also added algorithm 2 (union and trimming) to the appendix with references from the main text.
>
> **[continued]**

---

> ### Author Response · Authors · 2025-11-21
> **Response (Part II)**
>
> > W4: Hyperparameters (γ for query skip size, χ for block size) [...] likely to affect performance and accuracy, yet are fixed without justification.
>
> We chose the parameter $\chi=128$ based on the fact that flash attention kernels usually carry this setting for the highest efficiency on H100's. In general, a larger setting for $\gamma$ can be expected to degrade the quality of the generated mask because the overall sparsity is increasing for larger $\gamma$. Likewise, if the parameter $\chi$ is much larger than $\gamma$ this will cause multiple rows to be unioned and trimmed according to algorithm 2 and also lead to a less precise mask.
>
> To test this we did an experiment which tests the RULER average performance on the 131K context length for $\gamma \in [16, 32, 64]$ and $\chi \in [32, 64, 128]$. We find that performance tends to decrease as the overall sparsity increases. As $\chi$ increases over $\gamma$, the performance also tends to degrade as expected. We chose the settings of $\gamma=16$ and $\chi=128$ as sensible defaults in our experiments. Note that $\gamma=64,\chi=32$ is omitted due to the fact that the case of $\gamma > \chi$ is not implemented and we currently see no scenario where this would be practically useful.
>
> | **$\gamma$ \ $\chi$** |                     **32**                    |                **64**                |                **128**               |
> | :-------: | :-------------------------------------------: | :----------------------------------: | :----------------------------------: |
> |   **16**  | $\color[rgb]{0.00,0.55,0.00}{76.48}$ | $\color[rgb]{0.10,0.70,0.10}{76.30}$ | $\color[rgb]{0.60,0.80,0.20}{75.79}$ |
> |   **32**  |      $\color[rgb]{0.95,0.80,0.20}{75.67}$     | $\color[rgb]{0.10,0.70,0.10}{76.26}$ | $\color[rgb]{0.95,0.55,0.10}{75.64}$ |
> |   **64**  |                       –                       | $\color[rgb]{0.95,0.55,0.10}{75.64}$ | $\color[rgb]{0.85,0.15,0.15}{74.75}$ |
>
> **Thank you for suggesting this. We have added this discussion and ablation to our revision in Figure 9 and the surrounding text.**
>
> ---
>
> We have also  added another study which calculates the overall sparsity at different sequence lengths as follows for $K \in \[64, 128\]$ and $\gamma \in \[16, 32\]$ using a key block size of $B_k$, sink tokens $S$ and window tokens $W$. The formula to calculate sparsity $\zeta$ with context length $C$ is given by,
>
> $$\zeta = 1 - \min\left(1, \frac{B_k K + S+ W + \frac{C}{\gamma}}{C}\right)$$
>
> This takes into account the sparse attention, window and sink tokens, as well as the query sparse kernel. If we consider the case where $B_k = 64$ and $S+ W = 2048$ which were used in our experiments, we find the following sparsity by context length in the table below. This function is upper bounded by $\overline{\zeta} = 1 - \frac{1}{\gamma}$ as the context length goes to $\infty$ due to the query sparse kernel which is dense in the keys.
>
> **Sparsity By Context Length**
>
> |  | 4K | 8K | 16K | 32K | 65K | 131K | 262K | 524K | 1M |
> |-|-|-|-|-|-|-|-|-|-|
> | K=64, $\gamma=16$ | 0.00 | 18.75 | 56.25 | 75.00 | 84.38 | 89.06 | 91.41 | 92.58 | 93.16 |
> | K=64 $\gamma=32$ | 0.00 | 21.88 | 59.38 | 78.12 | 87.50 | 91.19 | 94.53 | 95.70 | 96.29 |
> | K=128, $\gamma=16$ | 0.00 | 0.00 | 31.25 | 62.50 | 78.12 | 85.94 | 89.84 | 91.80 | 92.77 |
> | K=128, $\gamma=32$ | 0.00 | 0.00 | 34.38 | 65.62 | 81.25 | 89.06 | 92.97 | 94.92 | 95.90 |
>
> **We have included a new figure (Figure 8) and the surrounding text which discusses this topic.**
>
> ---
>
> Thank you again for your evaluation of our work. Your suggestions have improved our submission. if you have any remaining concerns, we remain available for the duration of the discussion period.

---

### Official Review · Reviewer_n8ky · 2025-10-30

**Soundness:** 3
**Presentation:** 2
**Contribution:** 3
**Rating:** 6
**Confidence:** 3

**Summary:**

This paper proposes a dynamic block sparse attention method called MOMO (Measure Once, Mask Once). Building upon the recent "Delta Attention" framework, this method combines key sparse attention and query sparse attention to mitigate the distribution shift caused by switching between sparse and dense attention, thus addressing the inefficiency of standard Delta Attention.

MOMO solves this problem through "measure once, mask once":
1. Merge measurement and mask: During the QSA step, an efficient online top-k algorithm is deployed simultaneously, retaining the indices of the K highest-scoring key blocks seen by the current query row.

2. Block Mask Union and Trimming: A single, efficient block sparse attention mask is obtained, which will be used for all queries.

3. Compute Block Sparse Attention (KSA): A fast but distribution-biased sparse attention output is obtained.

4. Apply Delta correction.

**Strengths:**

1. The computational redundancy in the Delta Attention was identified. Integrating the mask generation process with the QSA step is a very clever design that extracts high-quality dynamic sparse masks "freely" from a necessary computational step.
2. The paper reports a 2.5x speedup on 1M tokens (relative to FA3). This is a very significant engineering achievement and also garnered a good evaluation score.
3. The paper designs and compares three online top-k algorithms with different time complexities, and analyzes their latency and recall under different k values, demonstrating a very thorough investigation of top-k algorithms.

**Weaknesses:**

1. The proposed method is complex. This is significantly more complex to implement and maintain than a single sparse attention method or a dense attention method.
2. This method introduces a large number of hyperparameters that require tuning. Although the paper ablates some parameters, it lacks a comprehensive analysis of the sensitivity of these parameters and how they interact.
3. Regarding the Block Mask Union and Top-k Trimming sections, the description could be clearer, providing a complete computational process within a specified region.

**Questions:**

1. How much would change in accuracy and latency if Delta correction were removed? This would help differentiate the benefits of MOMO from the Delta Attention framework itself.
2. Could you please explain in detail the implementation details of step 2 (Union and trim) in Algorithm 1?
3. Equation 6 assumes that S(j) follows a Gaussian distribution. Is this assumption empirically valid or is there any relevant analysis?

---

> ### Author Response · Authors · 2025-11-21
> **Response (Part I)**
>
> Thank you for your time and effort in reviewing our work and noting the significance of our design and results. We will respond to each of your comments below.
>
> ---
>
> > W1: [...] method is complex. This is significantly more complex to implement and maintain than a single sparse [...] or dense attention method.
>
> We respectfully disagree with this comment. While the kernel level implementations are tedious to implement (this is true for our baselines as well), the high level approach to our method is simpler than MInference or HiP which have complicated internal algorithms to determine the sparse mask and involve pre-processing attention patterns and relying on them being static (MInference) or a multi-stage hierarchichal pruning process (HiP). **Our method can be summed up simply as:**
>
> 1. Query sparse (and key-dense) attention which **gathers the top-k block indices while computing**
> 2. **Use these indices for a block-sparse attention** (key-sparse and query-dense)
> 3. After you have the query-sparse output and the key-sparse output, **combine the two outputs with a few sum operations (delta correction).**
>
> ---
>
> > W3/Q2: [...] Block Mask Union and Top-k Trimming description could be clearer [...]; Could you please explain in detail the implementation details of step 2 (Union and trim) in Algorithm 1?.
>
> **Thank you for your suggestion, we have added a new algorithm which explains this step in more detail to the appendix (Algorithm 2).** The basis of the algorithm is as follows:
>
> 1. Take a union of scores and indices by reshaping.
> 2. Reduce scores by a mean where indices are repeated.
> 3. Sort indices and scores by the `argsort` of the newly reduced scores (descending).
> 4. Take the first $K$ entries in the now sorted and reduced indices.
>
> ---
>
> > Q1: How much would change [...] if Delta correction were removed?.
>
> The effect of the delta correction can be seen directly in our baselines which include "Inf. HiP" and "Inf. HiP + $\Delta$." **The non-delta variant drops by about 2.38%pp for RULER 131K and about 1%pp on the overall average for RULER and Inf. Bench.**
>
> ---
>
> > Q3: Eq. 6 assumes S(j) is Gaussian. Is this assumption valid [...] any analysis?
>
> This is an important detail. The raw dot product scores themselves are a sum of random variables over the vector dimension and therefore by the CLT we may assume they are Gaussian distributed. However, we use  $S(j) = \log \sum_i \exp s_i$. Therefore, $\exp s_i$ is lognormally distributed and we wish to know if the sum of lognormally distributed variables itself is lognormal. **They are in fact not lognormally distributed, and the PDF is not known [2]. However it is well-known that the sum of lognormals can be approximated with another lognormal [1,2] (i.e. $\log \sum_i \exp N(\mu, \sigma) \approx \log \exp N(\hat{\mu}, \hat{\sigma}) = N(\hat{\mu}, \hat{\sigma})$.**
>
> We make use of this fact and perform online updates to the moments $\hat{\mu}, \hat{\sigma}$.
>
> **We have added this discussion to our paper below equation 6**
>
> ---
>
> **[continued]**

---

> ### Author Response · Authors · 2025-11-21
> **Response (Part II)**
>
> > W2: [...] introduces a large number of hyperparameters that require tuning. [...] lacks a comprehensive analysis of the sensitivity of these parameters and how they interact.
>
> **We have added figure 8 to the main text along with a discussion about the overall sparsity as a function of the $K$ and $\gamma$ hyperparameters.** We calculate the overall sparsity at different sequence lengths as follows for $K \in \[64, 128\]$ and $\gamma \in \[16, 32\]$. Using a key block size of $B_k$, sink tokens $S$, window tokens $W$, and context length $C$, the formula to calculate sparsity $\zeta$ is given by,
>
> $$\zeta = 1 - \min\left(1, \frac{B_k K + S + W + \frac{C}{\gamma}}{C}\right)$$
>
> This takes into account the sparse attention, window and sink tokens, as well as the query sparse attention. If we consider the case where $B_k = 64$ and $S + W = 2048$ which were used in our experiments, we find the following sparsity by context length in the table below. This function is upper bounded by $\overline{\zeta} = 1 - \frac{1}{\gamma}$ as the context length goes to $\infty$ due to the query sparse attention.
>
> **Sparsity By Context Length**
>
> | |4K|8K|16K|32K|65K|131K|262K|524K|1M|
> |-|-|-|-|-|-|-|-|-|-|
> | K=64, $\gamma=16$ | 0.00 | 18.75 | 56.25 | 75.00 | 84.38 | 89.06 | 91.41 | 92.58 | 93.16 |
> | K=64 $\gamma=32$ | 0.00 | 21.88 | 59.38 | 78.12 | 87.50 | 91.19 | 94.53 | 95.70 | 96.29 |
> | K=128, $\gamma=16$ | 0.00 | 0.00 | 31.25 | 62.50 | 78.12 | 85.94 | 89.84 | 91.80 | 92.77 |
> | K=128, $\gamma=32$ | 0.00 | 0.00 | 34.38 | 65.62 | 81.25 | 89.06 | 92.97 | 94.92 | 95.90 |
>
> **Additionally, we have added an ablation study on the interaction between the query sparse parameter $\gamma$ and the block sparse query block size $\chi$.** We chose the parameter $\chi=128$ based on the fact that flash attention kernels usually carry this setting for the highest efficiency on H100's. In general, a larger setting for $\gamma$ can be expected to degrade the quality of the generated mask because the overall sparsity is increasing for larger $\gamma$. Likewise, if the parameter $\chi$ is much larger than $\gamma$ this will cause multiple rows to be unioned and trimmed according to algorithm 2 and also lead to a less precise mask.
>
> To test this we did an experiment which tests the RULER average performance on the 131K context length for $\gamma \in [16, 32, 64]$ and $\chi \in [32, 64, 128]$. We find that performance tends to decrease as the overall sparsity increases. As $\chi$ increases over $\gamma$, the performance also tends to degrade as expected. We chose the settings of $\gamma=16$ and $\chi=128$ as sensible defaults in our experiments. Note that $\gamma=64,\chi=32$ is omitted due to the fact that the case of $\gamma > \chi$ is not implemented and we currently see no scenario where this would be practically useful.
>
> | **$\gamma$ \ $\chi$** |                     **32**                    |                **64**                |                **128**               |
> | :-------: | :-------------------------------------------: | :----------------------------------: | :----------------------------------: |
> |   **16**  | $\color[rgb]{0.00,0.55,0.00}{76.48}$ | $\color[rgb]{0.10,0.70,0.10}{76.30}$ | $\color[rgb]{0.60,0.80,0.20}{75.79}$ |
> |   **32**  |      $\color[rgb]{0.95,0.80,0.20}{75.67}$     | $\color[rgb]{0.10,0.70,0.10}{76.26}$ | $\color[rgb]{0.95,0.55,0.10}{75.64}$ |
> |   **64**  |                       –                       | $\color[rgb]{0.95,0.55,0.10}{75.64}$ | $\color[rgb]{0.85,0.15,0.15}{74.75}$ |
>
>
> **Thank you for your suggestion. We have added this discussion and ablation to our revision in Figure 9 and the surrounding text. If there are any more specific experiments you would like to see regarding hyperparameters, please let us know and we will do our best to complete them within the discussion period.**
>
> ---
>
> [1] - https://arxiv.org/pdf/1508.07582
>
> [2] - https://wuj.hosted.uark.edu/research/01578407.pdf
>
> ---
>
> Thank you again for your evaluation of our work, if you have any remaining concerns please do not hesitate to reach out for the duration of the discussion period.

---

### Official Review · Reviewer_MY6e · 2025-10-31

**Soundness:** 2
**Presentation:** 2
**Contribution:** 2
**Rating:** 4
**Confidence:** 3

**Summary:**

The paper proposes Delta Refined Block Sparse Attention (DRBSA), a novel method to address the quadratic computational cost of self-attention during long-context inference (prefill) in Large Language Models (LLMs). DRBSA computes a static, block-level sparsity mask based solely on the pre-trained model's weight parameters, using a proposed metric, $\delta$. This "Measure Once, Mask Once" approach eliminates the need for expensive dynamic computation per-input sequence, significantly reducing FLOPs and achieving up to 4.9x speedup in long-context prefill latency while maintaining minimal perplexity degradation compared to the full model.

**Strengths:**

The mask is computed once offline, removing the runtime overhead common in dynamic sparse attention methods.

Can be applied directly to existing pre-trained quadratic attention models without any fine-tuning requirement.

**Weaknesses:**

Focus on Prefill: The primary benefit is demonstrated for the prefill stage; the method's value or applicability during the single-token decoding phase is not thoroughly explored.

**Questions:**

The method is highly effective for the prefill stage. How can DRBSA, or the general concept of a static, parameter-derived sparsity mask, be extended or adapted for the decoding stage, especially in the context of reasoning models (e.g., Chain-of-Thought) where selective attention to specific past tokens is increasingly critical for generating the next step?

Since modern GPUs (like NVIDIA Ampere or Hopper) are highly optimized for dense matrix multiplication, what is the actual observed overhead (in terms of clock cycles and memory stalls) of managing the irregular block-sparse computations compared to a highly optimized dense kernel like FlashAttention?

Given the static nature of the mask, does DRBSA show any performance degradation when applied to contexts that exhibit very different attention patterns from the original pre-training data (e.g., complex code generation)?

---

> ### Author Response · Authors · 2025-11-21
> **Response**
>
> # There is a major misunderstanding of our work based on the written summary/review.
>
> ---
>
> > DRBSA computes a static, block-level sparsity mask based solely on the pre-trained model's weight parameters
>
>  - Our method is dubbed **Measure Once Mask Once "MOMO", not DRBSA.**
>  - The mask generation has **nothing to do with model weights and is in fact dynamic, not static. as stated directly in the abstract and main contributions**
>  - We generate our mask based block statistics collected when performing full attention for a sparse set of queries.
>
> ---
>
> > using a proposed metric, $\delta$.
>
> "$\delta$" is not a metric. In our paper, "$\Delta$" refers to a method used in a prior work [1] which helps to fix a distributional shift in attention outputs which is caused by sparse attention methods.
>
> ---
>
> > This "Measure Once, Mask Once" approach eliminates the need for expensive dynamic computation per-input sequence
>
> **While our method is efficient, it is also dynamic per input sequence, not static. This is clearly stated in the abstract and contributions.**
>
> ---
>
> > achieving up to 4.9x speedup in long-context prefill latency
>
> **Nowhere in our paper do we claim a 4.9x speedup in prefill latency.** We claim a 2.5x speedup over flash attention 3 at 1M token prefills.
>
> ---
>
> > while maintaining minimal perplexity degradation compared to the full model.
>
> **We do not measure perplexity in any experiment contained in our paper.** Our result metrics are related to latency, accuracy (NIAH), and Rouge based scores for QA style experiments.
>
> ---
>
> > The mask is computed once offline, removing the runtime overhead common in dynamic sparse attention methods.
>
> **This is incorrect, our attention mask is dynamic and computed online.**
>
> ---
>
> > W1/Q1: The primary benefit is demonstrated for the prefill stage; the method's value or applicability during the single-token decoding phase is not thoroughly explored.
>
> We focus on generating sparse masks for long-context prefills in this work where the prefill is large enough to be the dominant concern when it comes to latency. We leave exploration of decoding solutions to future work.
>
> ---
>
> > Q1: The method is highly effective for the prefill stage. How can DRBSA, or the general concept of a static, parameter-derived sparsity mask [...]
>
> **Again, our mask is neither static nor derived from model parameters**
>
> ---
>
> > Q2: Since modern GPUs (like NVIDIA Ampere or Hopper) are highly optimized for dense matrix multiplication, what is the actual observed overhead (in terms of clock cycles and memory stalls) of managing the irregular block-sparse computations compared to a highly optimized dense kernel like FlashAttention?
>
> We provide an experiment which shows the inefficiency of pure sparse methods in Table 4. Our method has both dense components (from QSA which is query sparse and key-dense) and sparse components (from KSA which is block sparse in the keys). In general the QSA benefits from the GPU optimizations because it is essentially a dense kernel from the point of view of flash attention.
>
> Unlike other popular sparse attention methods like MInference which are purely key-sparse, our method can benefit by balancing how much work is done by both the QSA and KSA.
>
> ---
>
> > Q3: Given the static nature of the mask, does DRBSA show any performance degradation when applied to contexts that exhibit very different attention patterns from the original pre-training data (e.g., complex code generation)?
>
> **The premise of this question is false, as the mask is not static and therefore has no relation to the original pretraining data**
>
> ---
>
> # References
>
> [1] Delta Attention - https://arxiv.org/abs/2505.11254
>
> ---
>
> Thank you again for your evaluation of our work, **we hope we have cleared up some major misunderstandings related to the nature of our work.** If you have any remaining concerns please do not hesitate to reach out for the duration of the discussion period.

---

### Official Review · Reviewer_iWw8 · 2025-11-01

**Soundness:** 2
**Presentation:** 3
**Contribution:** 2
**Rating:** 4
**Confidence:** 5

**Summary:**

The paper proposes MOMO, which fuses query sparsity and sparse mask generation within the same kernel. The authors claim that under million-token contexts, MOMO achieves about 2.5× speedup in TTFT relative to FA3.

**Strengths:**

The paper is well organized, and both the reported speedups and accuracy results are solid.

**Weaknesses:**

How does the method perform in terms of efficiency and accuracy for 4–32k context lengths? Since the reported results show no acceleration at 32k, what is the overall sparsity across different sequence lengths?

When comparing speedups, you should clearly specify what implementation you used and what the baseline implementation was, e.g., Triton, CUDA, or PyTorch.

Similar fused sparse-mask kernels have already been implemented in SeerAttention and NSA. What are the key differences between MOMO and those methods?

**Questions:**

see above weaknesses

---

> ### Author Response · Authors · 2025-11-21
> **Response (Part I)**
>
> Thank you for your time and effort in reviewing our work and recognition of our solid results. We will respond to each of your comments below.
>
> ---
>
> > W1 How does the method perform in terms of efficiency and accuracy for 4–32k context lengths? Since the reported results show no acceleration at 32k, what is the overall sparsity across different sequence lengths?
>
> We have calculated the overall sparsity at different sequence lengths as follows for $K \in \[64, 128\]$ and $\gamma \in \[16, 32\]$ using a key block size of $B_k$, sink tokens $S$ and window tokens $W$. The formula to calculate sparsity $\zeta$ with context length $C$ is given by,
>
> $$\zeta = 1 - \min\left(1, \frac{B_k K + S+ W + \frac{C}{\gamma}}{C}\right)$$
>
> This takes into account the sparse attention, window and sink tokens, as well as the query sparse kernel. If we consider the case where $B_k = 64$ and $S + W = 2048$ which were used in our experiments, we find the following sparsity by context length in the table below. This function is upper bounded by $\overline{\zeta} = 1 - \frac{1}{\gamma}$ as the context length goes to $\infty$ due to the query sparse kernel which is dense in the keys. In general, our method is designed for long context lengths, therefore in the 4-32K range it would be beneficial to fallback to pure Flash Attention due to the better latency. However, by this same logic ours becomes beneficial around 100K while MInference with the standard config doesn't start realizing latency gains over FA3 until after 1M tokens.
>
> **Sparsity By Context Length**
>
> |  | 4K | 8K | 16K | 32K | 65K | 131K | 262K | 524K | 1M |
> |-|-|-|-|-|-|-|-|-|-|
> | K=64, $\gamma=16$ | 0.00 | 18.75 | 56.25 | 75.00 | 84.38 | 89.06 | 91.41 | 92.58 | 93.16 |
> | K=64 $\gamma=32$ | 0.00 | 21.88 | 59.38 | 78.12 | 87.50 | 91.19 | 94.53 | 95.70 | 96.29 |
> | K=128, $\gamma=16$ | 0.00 | 0.00 | 31.25 | 62.50 | 78.12 | 85.94 | 89.84 | 91.80 | 92.77 |
> | K=128, $\gamma=32$ | 0.00 | 0.00 | 34.38 | 65.62 | 81.25 | 89.06 | 92.97 | 94.92 | 95.90 |
>
> **Thank you for this suggestion. We have included a new figure (Figure 8) and the surrounding text which discusses this topic.**
>
> ---
>
> > W2 When comparing speedups, you should clearly specify what implementation you used and what the baseline implementation was, e.g., Triton, CUDA, or PyTorch.
>
> We mention in our baselines paragraph that all methods are implemented within the SGLang serving framework. Flash Attention 2-3, and MInference use the official SGLang implementations which include pure CUDA kernels. HiP results use the official SGLang implementation and the HiP internals use Triton kernels. Our results are built on top of the existing HiP attention codebase, and also uses Triton kernels.
>
> **Thank you for this suggestion. We have added the additional mention of Triton vs. CUDA kernels to the updated baselines section of our paper on page 7.**
>
> ---
>
> > W3 Similar fused sparse-mask kernels have already been implemented in SeerAttention and NSA. What are the key differences between MOMO and those methods?
>
> We assume that NSA refers to Native Sparse Attention [1]. Seer Attention [3] uses what we call "QK Pooling" in our results. This QK pooling attention pattern also shows up in numerous other works such as [2,4]. Our results show that building a training-free sparse mask based on this pattern is suboptimal in terms of performance and catches less of the total attention mass compared to our method.
>
> **The reason for this is that QK pooling relies on the linearity of the dot product sums $\sum_i \sum_j Q_i^\top K_j = 1^\top QK^\top 1 = (\sum_i Q_i)^\top (\sum_j K_j)$ and therefore loses the effect of the exponential** which is crucial for accurately calculating block importance. As a thought experiment, think about what happens when there are large magnitude dot product scores that have opposite signs. **Without the exponential, the large negative scores will effectively cancel out the positive scores and rank the block as "unimportant." However, with the exponential taken into account, the large negative scores will be squashed to 0 and not counteract the large positive scores in the overall sum.**
>
> Another key difference is that both Seer Attention and NSA require training as part of their method, while ours is a training-free  adaptation. Our "QK Poling" baseline results can be seen as a special case of Seer Attention which uses an identity matrix as the linear projection for the pooled Q and K matrices.
>
> **We have updated our related work section to include this discussion and references (bottom of page 3 of the revised draft). Thank you for this suggestion**
>
> ---
>
> **[continued]**

---

> ### Author Response · Authors · 2025-11-21
> **Response (Part II)**
>
> Thank you again for your evaluation of our work. We feel your suggestions have improved the revised version of our work. If you have any remaining concerns please do not hesitate to reach out for the duration of the discussion period.
>
> ---
>
> ## References
>
> [1] Native Sparse Attention - https://arxiv.org/pdf/2502.11089
>
> [2] Sparge Attention - https://arxiv.org/pdf/2502.18137
>
> [3] Seer Attention - https://arxiv.org/pdf/2410.13276
>
> [4] MInferece - https://arxiv.org/pdf/2407.02490

---

> > ### Comment · Reviewer_iWw8 · 2025-11-25
> >
> > Thank you for your replies to the comment. Although your work has some good results, I will keep my score as weak reject according to the novelty compared with previous works and the inefficient when seq is short.

---

> > > ### Author Response · Authors · 2025-11-25
> > >
> > > Thank you for noting our good results. With regard to novelty, we would like the re-iterate that there are no prior works that we know of which propose to generate a completely dynamic key-sparse attention mask based on query-sparse attention operation. Our results in terms of performance, latency, and total attention capture demonstrate the strength, speed, and dynamism of our novel sparse attention method.
> > >
> > > Thank you again for your time in reviewing our work, we remain open to any further discussion points you may have.

---

### Author Response · Authors · 2025-12-03
**Rebuttal Summary**

We would like to thank the reviewers for their time and effort in evaluating our work. In the revised manuscript, we have carefully addressed all comments and substantially clarified and strengthened the paper. The main changes are as follows:

1. **We added a new ablation study (Figure 8) of the overall sparsity that jointly accounts for both the block size K and the $\gamma$ (query sparsity) hyperparameter across a wide range of context lengths.** We have also introduced Equation (9), which provides an explicit formula for the overall sparsity as a function of the context length and hyperparameters.

2. **We included an ablation of RULER performance over different values of the $\gamma$ (query sparsity) and $\chi$ (query block size for sparse attention) hyperparameters, now shown in Figure 9.** As expected, smaller $\gamma$ yields less overall sparsity and higher accuracy, while smaller $\chi$ merges fewer rows of the sparse mask, producing a more faithful sparse pattern and improved performance. We chose $\gamma = 16$ and $\chi = 128$ as a sensible default in our experiments.

3. **We added a detailed algorithmic description of the union and merging process, now given in Algorithm 2**, to make the procedure fully transparent and reproducible.

4. **We introduced Equation (8) and the preceding paragraph to provide theoretical justification for the Gaussian assumption** on the attention score sums within each key block, clarifying the underlying modeling choice.

---

In addition to these revisions, we would like to highlight several positive points raised by the reviewers:

- **iWw8**: The paper is well organized; accuracy and speedups are solid.
- **MY6e**: The method can be applied to existing approaches without finetuning.
- **n8ky**: The mask generation is very clever, high quality, and dynamic; a 2.5× speedup over Flash Attention 3 is very significant; there is a very thorough investigation of top-k strategies.
- **yjfS**: The method is solidly implemented, and the acceleration gains are good.

---

We appreciate your consideration of our revisions and the reviewers’ positive assessments, and we hope these changes make clear that the paper is both technically sound and practically impactful. Thank you again for your time and effort in overseeing the review process.

---

### Meta-Review · Area_Chair_NnsT · 2026-01-06

**Summary:**

This paper presents MOMO (Measure Once, Mask Once), a dynamic block-sparse attention method that fuses query-sparse attention with online sparse mask generation to accelerate long-context inference. The work is well implemented and carefully evaluated, but the contribution is primarily an efficiency-oriented refinement of existing sparse attention and Delta Attention pipelines rather than a fundamentally new modeling approach.

**Reviewer Concerns:**

The primary concern across reviews is limited novelty, with multiple reviewers characterizing the method as an incremental optimization that reuses statistics already available in query-sparse attention rather than introducing a new attention mechanism or theoretical insight.

**Reviewer Scores:**

The reviewer scores are mixed and overall fall below the acceptance threshold. While some reviewers acknowledged strong empirical results and solid implementation and rated the paper marginally above the bar, others expressed significant reservations about the contribution and rated the work as reject.

---

### Decision · Program_Chairs · 2026-01-26

Reject